# Ocelots in the moonlight: Influence of lunar phase on habitat selection and movement of two sympatric felids

**Maksim Sergeyev**[1¤a]*, **Jason V. Lombardi**[1¤b], **Michael E. Tewes**[1], **Tyler A. Campbell**[2]

**1** Caesar Kleberg Wildlife Research Institute, Texas A&M University Kingsville, Kingsville, TX, United States of America, **2** East Foundation, San Antonio, TX, United States of America

¤a Current address: Colorado State University, Fort Collins, CO, United States of America
¤b Current address: Wildlife Health Laboratory, California Department of Fish and Wildlife, Rancho Cordova, CA, United States of America
* ecomakismsergeyev@gmail.com

## Abstract

Various landscape and environmental factors influence animal movement and habitat selection. Lunar illumination affects nocturnal visual perception of many species and, consequently, may influence animal activity and habitat selection. However, the effects of varying moon stage may differ across taxa. Prey species often reduce activity during highly visible periods of night while predators may increase activity or alter their habitat use. Ocelots (*Leopardus pardalis*) and bobcats (*Lynx rufus*), two nocturnal predatory felids that coexist in southern Texas, may also alter their behavior in response to the phase of the moon. To evaluate the effects of lunar phase on habitat selection of ocelots and bobcats, we executed a step selection analysis using high-frequency GPS-telemetry data collected on each species (ocelot, N = 8; bobcat, N = 13) in southern Texas during 2017–2021 and compared step length during new versus full moons. We predicted that ocelots would increase use of dense thornshrub to reduce their visibility during a full moon. However, as bobcats are habitat generalists and are more active during crepuscular periods, we predicted less influence of moon phase on activity. Ocelots did not alter habitat selection in response to lunar phase but moved shorter distances during full moon phases. Conversely, bobcats selected for greater vegetation cover during full moons, possibly to facilitate hunting during brighter periods, but exhibited no difference in movement across lunar phase. We provide, to our knowledge, the first example of habitat selection by predators in relation to lunar phase and show differences across new versus full moons by ocelots and bobcats such that ocelots alter step length but not habitat selection while bobcats altered habitat selection but not step length in response to shifting lunar phase. Further, we suggest the high potential for ocelot-vehicle collisions on darker nights due to increased movement by ocelots and poor visibility for drivers.

**Data Availability Statement:** Any R code and bobcat GPS data used in the analysis available in Github repository: https://github.com/MaksimSergeyev/LunarPhasePLOSOneData-Code.

**Funding:** Cooperative funding was provided by East Foundation, U.S. Fish and Wildlife Service, the Brown Foundation, the Tim and Karen Hixon Foundation and the Feline Research Program at the Caesar Kleberg Wildlife Research Institute. Funders had no role in study design, data collection and analysis, or decision to publish with the exception of the East Foundation which played a role in data collection, manuscript review and decision to publish.

**Competing interests:** The authors have declared that no competing interests exist.

## Introduction

Movement and habitat use of animals can depend on a wide range of biotic and abiotic factors across temporal scales [1–3]. For nocturnal animals, their activity patterns can be shaped by night length and temperature, as well as moon phase and luminosity [4, 5]. The moon impacts terrestrial processes through gravitational effects, tidal shifts, and lunar illumination, all of which vary throughout the lunar cycle [6]. Effects of varying lunar phases on animal behavior have been described for various mammals, birds, and insects [4, 5]. Influence of lunar stage may occur at larger scales, as in the case of population cycles [7] or may result in fine-scale differences in behavior of individuals [5]. The influence of varying lunar phase differs across taxa and may have differing impacts on predators versus prey species.

Lunar illumination impacts the detection ability of visually oriented species, both predators and prey [8] and may influence foraging success and risk of predation [9]. Changes in lunar illumination can have differing effects across species and can be moderated by visual acuity of the species and habitat cover [10]. Prey species may reduce activity levels during periods of heightened predation risk [11]. At night, greater illumination during a full moon is associated with an increased risk of predation for small mammals while the darkness during a new moon creates a reduced risk of predation [12]. As a result, many prey species alter their behavior during full moons and will typically reduce activity and alter habitat use to minimize vulnerability [5]. White-footed mice (*peromyscus leucopus*) foraged more selectively and abandoned foraging more readily during full moons [13]. Cricetid rodents (*Phyllotis darwini*) reduced total food consumption during full moons [8]. Snowshoe hares (*Lepus americanus*) decreased distances moved during brighter nights [11]. While predation risk increases with visibility, it is also worth noting that foraging efficiency may increase, as well as ability for prey to detect predators [14]. As a result, predators may alter their behavior in response to changes in activity and habitat selection of prey.

The behavior of predators is often driven by the behavior of prey species [15], therefore, as prey alter their activity around lunar phases, predators may adjust their behavior accordingly. In response to decreased activity of prey species, predators may respond by either increasing effort expended on searching for prey or reduce distance traveled to conserve energy and increase efficiency [16]. Red foxes (*Vulpes vulpes*) and maned wolves (*Chrysocyon brachyurus*) often reduce movement during full moons, likely to conserve energy during periods of less prey availability [16, 17]. Conversely, predators may also increase activity during new moons when visibility is lowest. Iberian lynx (*Lynx pardinus*) reduced overall movement but concentrated hunting efforts on areas with greater prey density during new moons [17]. Attacks on humans (*Homo sapiens*) by African lions (*Panthera leo*) increased 2–4 times after a full moon, when illumination was lower, compared to the period preceding a full moon [18]. The increased luminosity of full moons may also facilitate higher levels of increased nocturnal activity for less nocturnal species, as in the case of cheetahs (*Acinonyx jubatus*) and African wild dogs (*Lycaon pictus*), leading to increased activity during brighter periods [19, 20]. Predators may hunt more successfully due to increased visibility on brighter nights [21], however, the likelihood of detection by prey also increases [14]. To offset increased detectability by prey, predators may alter habitat selection, for example increasing use of vegetation cover to aid in camouflage or ambush hunting strategies.

The ocelot (*Leopardus pardalis*) is a medium-sized felid native throughout Central and South America, as well as the southern tip of Texas in the United States [22, 23]. Ocelots are strongly associated with dense vegetation cover across their geographic range [22, 24, 25]. Ocelots are nocturnal [22, 26], and generally rest during the day and hunt and patrol territory at night [26]. Further, their favored prey species (small rodents) have been found to alter use of

microhabitats in response to changing lunar phases [27]. As such, ocelots may respond by altering movement and habitat selection in response to changing illumination. To remain camouflaged and facilitate hunting strategies, ocelots may further increase use of dense vegetation during brighter stages of the moon. Like other felids, bobcats (*Lynx rufus*) hunt at night, however, their eyes are smaller and less adapted to nocturnal vision than other nocturnal felines [12, 28]. Bobcats exhibited greater diurnal and crepuscular activity compared to ocelots [26] and utilized a wider range of land cover types [26, 29]. As such, bobcats may exhibit less response to changing lunar phases than a strongly nocturnal species with a narrower habitat niche such as the ocelot [26].

For nocturnal species, conditions at night can be greatly influenced by the phase of the moon. Visibility can greatly change between new and full moons, and with it the behavior of wildlife. Felids frequently hunt at night and, as such, may be greatly influenced by varying luminosity across moon phases [18]. Studies on the effects of lunar phase on wildlife are limited and have largely focused on prey species [30, 31]. Less is known about the behavior of nocturnal predators in response to shifting lunar phase. We examined differences in daily movement and habitat selection of ocelots and bobcats in response to varying lunar phase. If varying lunar phases create ecologically meaningful differences in visibility or prey activity, ocelots and bobcats will respond by altering movement and habitat selection. We predicted that ocelots would increase use of dense vegetation cover during brighter periods and decrease movement during full moons. We predicted less effect of shifting lunar phase on the activity and habitat selection of bobcats as they are more active during diurnal and crepuscular periods and use a broader range of cover types than ocelots. Understanding the factors that influence activity and habitat use will improve overall understanding of the ecology of ocelots and foraging ecology of the species. By identifying differences in habitat selection between ocelots and bobcats, we can better understand how coexistence is facilitated and, ultimately, improve conservation of a threatened, native carnivore. In addition, understanding the factors that influence nocturnal movement can potentially improve recovery and conservation efforts to minimize the impact of vehicle collisions, the largest source of mortality for Texas ocelots [32, 33].

## Methods

### Study area

We conducted this study on the East Foundation's El Sauz Ranch and the Yturria Family's San Francisco Ranch located in Willacy and Kenedy counties in the southern tip of Texas, USA (Fig 1). The El Sauz Ranch (113 km$^2$) is an operational cattle (*Bos taurus indicus*) ranch with an emphasis on jointly managing land for native wildlife. The ranch features a variety of landscape features ranging from areas of woody vegetation cover, grasslands, prairies, sand dunes, coastal wetlands, and anthropogenic water features. The San Francisco Ranch (25.9 km$^2$) manages land for hunting of ungulates, land stewardship, and the conservation of ocelots [26]. This ranch also features two conservation easements (totaling 1.98 km$^2$), managed by the US Fish and Wildlife Service Lower Rio Grande Valley National Wildlife Refuge Complex [26], which contain highly dense patches of woody and herbaceous vegetation. Patches of woody vegetation in these area are comprised of crucifixion thorn (*Castela emoryi*), live oak (*Quercus virginiaI*), lotebush (*Ziziphus obtusifola*), desert olive (*Forestiera angustifolia*), snake-eyes (*Phaulothamnus spinescens*), honey mesquite (*Prosopis glandulosa*), white brush (*Aloysia gratissima*), lime prickly ash (*Zanthoxylum fagara*), crucita (*Chromolaena odorata*), spiny hackberry (*Celtis pallida*), and thornshrub (*Acacia farnesiana*) [23, 25, 26].

The region has a subtropical and semi-arid climate and temperatures in the area typically range from 10˚C to 36˚C; mean annual temperature is 23˚C [34, 35]. Woody vegetation in the

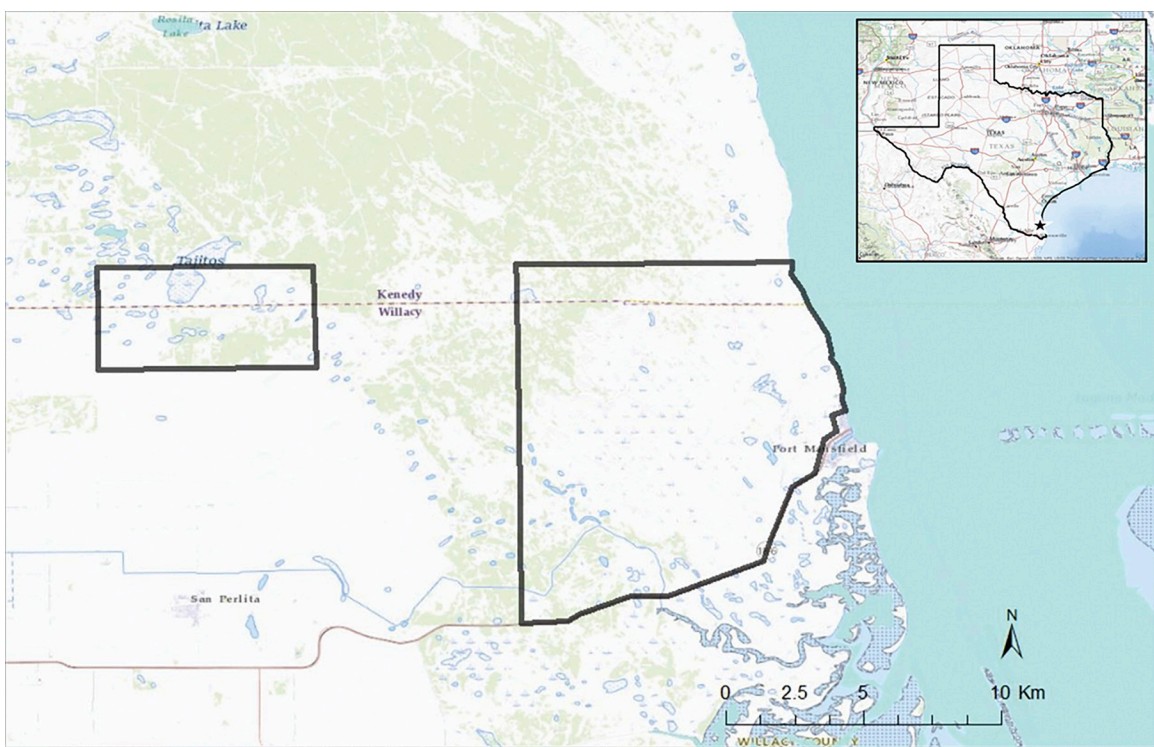

**Fig 1. Study area for examining habitat selection and movement rates of ocelots (*Leopardus pardalis*) and bobcats (*Lyn rufus*) during new and full moons from 2017 to 2021.** The left polygon denotes the Yturria Family's San Francisco Ranch, and the right polygon denotes the East Foundation's El Sauz Ranch in Willacy and Kenedy counties, Texas, USA. Figure was created in ESRI ArcMap 10.8.1.

area forms patches of dense canopy that can provide some reprieve from the extreme heat during the day. The area receives inconsistent rainfall leading to periodic drought; mean annual rainfall is 68 cm with seasonal variation, typically greater in the spring [34, 35]. This area has very low light pollution and is considered Bortle class 3, rural sky [36].

## Animal captures

We captured 8 ocelots (3 males and 5 females) and 13 bobcats (7 males and 6 females) between January 2017 and May 2021. All captured animals were of adult age. We used single-door Tomahawk box-traps (108 x 55 x 40cm; Tomahawk Trap Co., Tomahawk, WI, USA). We baited the box-traps with a live chicken (*Gallus gallus*) or pigeon (*Columbia livia*) contained in a separate compartment inaccessible to the trapped animal. Captured ocelots and bobcats were immobilized using a 4:1 mixture of tiletamine hydrochloride and zolazepam hydrochloride (Telazol™, Zoetis, Florham Park, NJ, USA) at a dose of 5mg/kg in 2017 [37] and a mixture of ketamine hydrochloride (4–5 mg/kg) and medetomidine HCl (0.05 mg/kg) and received a reversal of 5 mg of atipamezole per 1 mg medetomidine (ZooPharm, Laramie, WY, USA) from 2019–2021 [23, 26]. Each individual was then fitted with a Lotek Minitrack GPS or Lite-track satellite Iridium-GPS collar (Lotek™, New Market, ON, Canada). Collars recorded locations every 30–60 minutes and were programmed to automatically drop after either 4–6 month period or a 12-month period. Capture and handling of wildlife were conducted in accordance with United States Fish and Wildlife Service permit (#PRT-676811), Texas Parks and Wildlife Department permit (#SP0190-600), and Texas A&M University Kingsville Institutional Animal Care and Use Committee protocols (2012-12-20B-A2, 2019-2-28A-2-28B).

### Analysis

**Habitat selection.** We compared habitat selection during new moons versus full moons. As we were interested in nocturnal behavior, we removed locations collected during the day (sunrise–sunset) and conducted analyses on nighttime locations (sunset–sunrise). Lunar phase of each GPS location was classified using the 'lunar.phase' function in the 'lunar' package in R (version 4.1.2) [38, 39]. We used the 4-phase classification (new, full, waxing, and waning) and retained only locations collected during the new and full moons. We determined habitat selection using a step selection framework and created 10 random steps for each observed step, based on the step length and turning angle between successive steps [40]. We obtained landscape characteristics using light image detection and ranging (LiDAR) data flown by the United States Geological Survey in 2018 [41]. LiDAR data were classified using LP360 by GeoCue and landscape rasters were created at a spatial resolution of 10 m [26]. Data were classified into one of the following classes: water, road, buildings, ground, and vegetation. To understand how use of landscape characteristics differed between lunar phases, we considered the following landscape variables: % vertical canopy cover, distance to dense cover ($> 75\%$ canopy; m), distance to open areas ($< 25\%$ canopy; m), patch area of dense cover ($m^2$) and density of 0–1 m vegetation (points per cell; 26). We selected these variables to characterize the horizontal and vertical structure of dense vegetation and open, non-wooded areas characterized by bare ground and open herbaceous cover. We modeled habitat selection using mixed-effect logistic regression models in the 'survival' package in R [38–40, 42]. We evaluated a linear model that included the five landscape variables described and a random intercept of animal ID [40] separately for data collected during new moons versus full moons for ocelots and bobcats and compared selection between lunar phases and between species.

**Step length.** We evaluated differences in step length between new moons and full moons using an analysis of variance (ANOVA). In addition, we examined the effect of nocturnal conditions on the step length of ocelots and bobcats. Specifically, we examined the effect of changing illumination and night length on step length. Illumination was obtained for each GPS location using the 'lunar.illumination' function in the 'lunar' package in R [38, 39]. We obtained the length of night by using the 'daylength' function in the 'geosphere' package in R [43] and subtracting this value from 24 (the total diel period). We modeled step length of ocelots and bobcats as a function of night length and illumination using mixed effect linear regression models and the 'lmer' function in the 'lme4' package in R [39, 44].

## Results

### Habitat selection—Ocelots

We collected a total of 4,567 and 4,101 GPS locations from ocelots during the full and new moon respectively. Habitat selection of ocelots was highly similar between new and full moons (Table 1; Fig 2). Across lunar phase, ocelots selected closer to dense cover ($> 75\%$ canopy cover; OR = 0.58, 95% CI = [0.44, 0.76] during full moons and 0.56, 95% CI = [0.39, 0.79] during new moons), greater vertical canopy cover (OR = 1.78, 95% CI = [1.53, 2.06] during full moons and 1.61, 95% CI = [1.38, 1.88] during new moons), and higher density of 0–1 m vegetation (OR = 1.49, 95% CI = 1.29, 1.71 during full moons and 1.36, 95% CI = [1.26, 1.47 during new moons). Ocelots selected larger patches of dense cover during new moons (OR = 1.14, 95% CI = [0.99, 1.31]; $\alpha = 0.07$) but used larger patches in proportion to availability during full moons (OR = 1.04, 95% CI = [0.94, 1.16]).

### Habitat selection–Bobcats

We collected a total of 7,132 and 7,158 GPS locations from bobcats during the full and new moon respectively. Habitat selection of bobcats differed between new and full moons (Table 2;

**Table 1. Habitat selection of ocelots (*Leopardus pardalis*) during new and full moons in southern Texas, USA from 2017–2021, modeled using mixed-effect logistic regression models.** Models included distance to open areas (< 25% canopy cover; 'DistLowCover'), distance to dense cover (> 75% canopy cover; 'DistHeavyCover'), patch area of dense cover ('PatchArea'), percent vertical canopy cover ('CanopyCover') and density of 0–1 m vegetation (Density1mVeg). Each model included animal ID as a random intercept; all variables were scaled and centered by standard deviation.

| New Moon | Coef | exp(Coef) | Std. Error(coef) | p Value |
|---|---|---|---|---|
| DistLowCover | 0.0911 | 1.0954 | 0.0250 | 0.3013 |
| DistHeavyCover | -0.5851 | 0.5571 | 0.1454 | 0.0009 |
| PatchArea | 0.1303 | 1.1392 | 0.0598 | 0.0710 |
| CanopyCover | 0.4758 | 1.6092 | 0.0413 | < 0.0001 |
| Density1mVeg | 0.3060 | 1.3580 | 0.0183 | < 0.0001 |
| **Full Moon** | | | | |
| DistLowCover | 0.0847 | 1.0883 | 0.0241 | 0.4170 |
| DistHeavyCover | -0.5522 | 0.5757 | 0.1552 | 0.0001 |
| PatchArea | 0.0443 | 1.0453 | 0.0336 | 0.3917 |
| CanopyCover | 0.5752 | 1.7775 | 0.0417 | < 0.0001 |
| Density1mVeg | 0.3977 | 1.4884 | 0.0168 | < 0.0001 |

Fig 2). During full moons, bobcats selected closer to dense cover (OR = 0.73, 95% CI = [0.60, 0.87]) and larger patches of dense cover (OR = 1.12, 95% CI = [1.08, 1.17]), however, during new moons bobcats used these areas in proportion to availability (OR = 0.84, 95% CI = [0.68, 1.04] for distance to dense cover and 1.03, 95% CI = [0.93, 1.14] for patch area of dense cover). Across lunar phase, bobcats selected closer to open areas (< 25% canopy cover; OR = 0.89, 95% CI = [0.81, 0.98] during full moons and 0.89, 95% CI = [0.83, 0.96] during new moons), greater vertical canopy cover (OR = 1.11, 95% CI = [1.00, 1.25] during full moons; α = 0.06 and 1.17, 95% CI = [1.06, 1.30] during new moons), and greater density of 0–1 m vegetation (OR = 1.10, 95% CI = [1.05, 1.14] during full moons and 1.11, 95% CI = [1.06, 1.15] during new moons).

## Step length–Ocelots & bobcats

Ocelots moved greater distances during new moons compared to full moons (F value = 5.36, p = 0.021; mean step length between 30 min locations = 122.83 m during full moons and 131.57 during new moons; Fig 3). Step length of ocelots was positively associated with length of night, such that each increasing hour of night was associated with an average of 10.32 m greater step length ocelots (Table 3). Increasing illumination was negatively associated with movement, such that each unit increase in illumination was associated with 11.26 m shorter step length by ocelots.

Step length of bobcats did not differ between new moons and full moons (F value = 2.65, p = 0.104; mean step length between 30 min locations = 120.24 m during full moons and 124.08 during new moons). Bobcats moved greater distances during longer nights, such that each hour increase in the length of night was associated with 10.58 m greater step length by bobcats. Varying illumination did not affect the step length of bobcats.

## Discussion

Habitat selection and movement of wildlife are influenced by a variety of abiotic factors [45]. Among these, changing lunar phase has the potential to influence habitat selection of predators and prey species [9, 14]. Prey species may reduce movement or select for more secure habitat during brighter periods, the predation risk hypothesis, or alternatively may increase movement due to increased foraging success and predator detection, the visual acuity hypothesis [46].

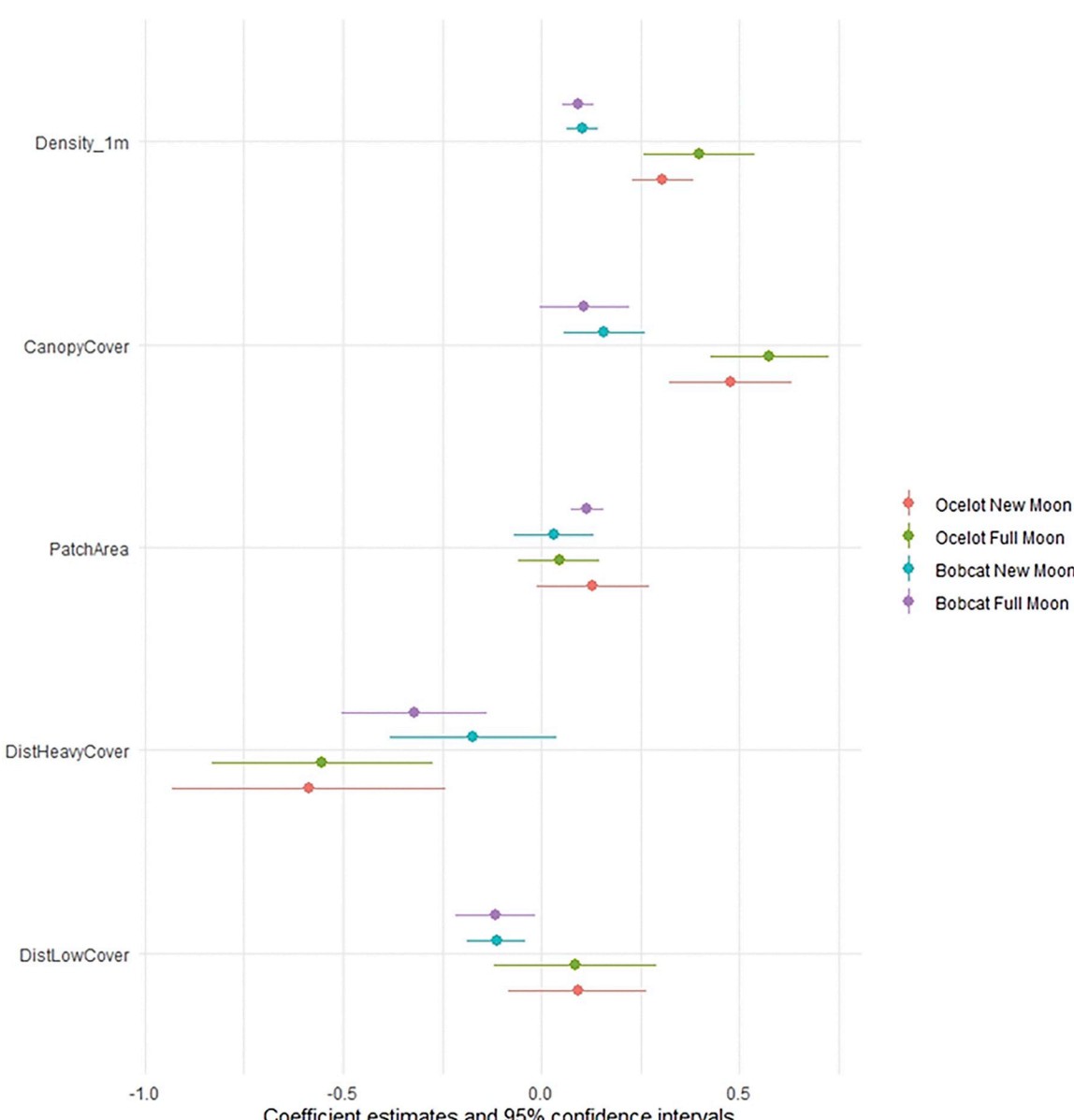

**Fig 2. Coefficient estimates from model of habitat selection of ocelots (*Leopardus pardalis*) and bobcats (*Lynx rufus*) during new and full moons in southern Texas, USA from 2017–2021.** We used mixed effect logistic regression models that included distance to open areas (< 25% canopy cover; 'DistLowCover'), distance to dense cover (> 75% canopy cover; 'DistHeavyCover'), patch area of dense cover ('PatchArea'), percent vertical canopy cover ('CanopyCover') and density of 0–1 m vegetation (Density1mVeg). Each model included animal ID as a random intercept; all variables were scaled and centered by standard deviation.

Predators may similarly adjust their movement and habitat selection in response to changing prey behavior [15], however, habitat selection of predators in response to changing lunar phase remains largely undocumented. We examined habitat selection and movement of ocelots and bobcats during new and full moons and observed differences between species and lunar phase. Further, our findings reveal new insights that add to a growing literature regarding the biology, coexistence and interactions of these sympatric felids in parts of their shared geographic range.

**Table 2. Habitat selection of bobcats (*Lynx rufus*) during new and full moons in southern Texas, USA from 2017–2021, modeled using mixed-effect logistic regression models.** Models included distance to open areas (< 25% canopy cover; 'DistLowCover'), distance to dense cover (> 75% canopy cover; 'DistHeavyCover'), patch area of dense cover ('PatchArea'), percent vertical canopy cover ('CanopyCover') and density of 0–1 m vegetation (Density1mVeg). Each model included animal ID as a random intercept; all variables were scaled and centered by standard deviation.

| New Moon | Coef | exp(Coef) | Std. Error(coef) | p Value |
|---|---|---|---|---|
| DistLowCover | -0.1132 | 0.8930 | 0.0209 | 0.0022 |
| DistHeavyCover | -0.1713 | 0.8426 | 0.0316 | 0.1096 |
| PatchArea | 0.0332 | 1.0338 | 0.0246 | 0.5170 |
| CanopyCover | 0.1577 | 1.1708 | 0.0219 | 0.0025 |
| Density1mVeg | 0.1032 | 1.1087 | 0.0133 | < 0.0001 |
| **Full Moon** | | | | |
| DistLowCover | -0.1157 | 0.8907 | 0.0208 | 0.0228 |
| DistHeavyCover | -0.3196 | 0.7264 | 0.0383 | 0.0005 |
| PatchArea | 0.1148 | 1.1216 | 0.0246 | < 0.0001 |
| CanopyCover | 0.1084 | 1.1145 | 0.0225 | 0.0602 |
| Density1mVeg | 0.0932 | 1.0977 | 0.0136 | < 0.0001 |

Contrary to our first prediction, habitat selection of ocelots was highly consistent between lunar phases. With the exception of selecting for larger patches during new moons, habitat selection was nearly identical between new and full moons. Selection for larger patches during new moons may be potentially explained by differences in habitat selection of prey, wherein

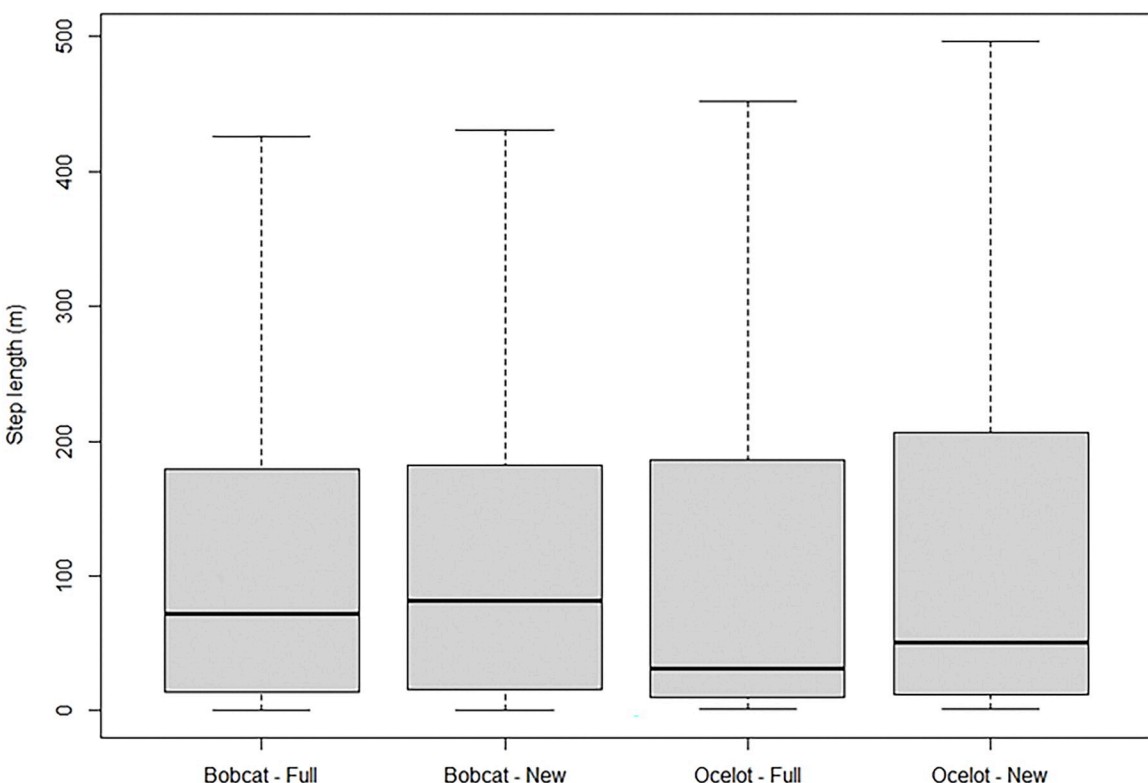

**Fig 3. Step length (m) of ocelots (*Leopardus pardalis*) and bobcats (*Lynx rufus*) during new and full moons in southern Texas, USA from 2017–2021.** Boxes represent the range from the lower quartile to the upper quartile; horizontal black line denotes the median value. Whiskers represent 1.5X of the interquartile range. Step length differed between new and full moons for ocelots but not bobcats, wherein ocelots moved greater distances during new moons compared to full moons.

**Table 3. Estimates from mixed effect linear model describing step length as a function of length of night ('Night-Length') and lunar illumination ('Illumination') for ocelots (*Leopardus pardalis*; top) and bobcats (*Lynx rufus*; bottom) in South Texas, USA from 2017–2021.**

| Ocelots | | | |
|---|---|---|---|
| | Estimate | Std. Error | p Value |
| Intercept | 25.9080 | 40.6690 | 0.5318 |
| NightLength | 10.3190 | 2.2070 | < 0.0001 |
| Illumination | -11.2590 | 3.8610 | 0.0036 |
| **Bobcats** | | | |
| | Estimate | Std. Error | p Value |
| Intercept | 0.5666 | 22.9948 | 0.980 |
| NightLength | 10.5778 | 1.6018 | < 0.001 |
| Illumination | -3.2758 | 2.4766 | 0.1860 |

prey species may remain within larger patches during darker nights due to poor foraging efficiency and in response, ocelots may be increasing use of these larger patches of dense vegetation. Ocelots have been documented altering their habitat selection because of increased lunar illumination, avoiding open areas and increasing use of dense vegetation during brighter lunar periods [22, 27], however, we did not document such a relationship. These differences may be attributable to an attempt to reduce overlap with larger, more dominant felids such as jaguar (*Panthera onca*) and puma (*Puma concolor*; absent in our study area), which did not alter their movement during moon phases [47]. Further, ocelots have strong eyesight and are adept at nocturnal hunting, such that they may be able to effectively hunt regardless of the presence or absence of light on the landscape [14, 28, 48]. We did, however, observe differences in movement such that ocelots moved less during full moons than new moons and had shorter step lengths with increasing illumination, partially supporting our first prediction. Past literature has shown contrasting results on the activity of ocelots in relation to lunar phase. Ocelots showed the greatest movement during darker nights surrounding the new moon [22, 49], consistent with our results. In other locations, however, movement of ocelots was unrelated to lunar phase [50–52] or increased during brighter periods [14], contrasting our results. The differences observed in these studies may be attributable to some combination of differences in the vegetation communities, presence of more dominant felids, or activity of prey in response to the lunar phase. Activity of ocelots and their prey species overlapped in relation to lunar phase and any differences in activity of prey were thought to relate to differences in visual ability of the species [14, 51]. A similar pattern of reduced movement during full moons was observed in maned wolves, also attributed to temporal differences in prey availability [16]. Data on preferred prey within our study area were not collected during the duration of our study, however, we may expect similar overlap between ocelots and their prey species which may suggest greater activity during darker nights and explain the greater movement rate of ocelots during new moons.

We found partial support for our second prediction. Bobcats exhibited differences in habitat selection between new and full moons, in that during full moons, bobcats selected for larger patches of cover and selected closer to dense cover. This difference in habitat selection may be due to activity of prey or related to reducing detection while hunting. We also observed some trends that appeared in contrast to the behavior of ocelots, wherein ocelots selected larger patches of cover during new moons while bobcats used cover at a greater extent during full moons. While we were unable to find a comparison for habitat selection of bobcats in relation to lunar phase, bobcats and ocelots did exhibit contrasting activity patterns regarding

movement wherein ocelots were most active during new moons, consistent with our results, while bobcats were more active during the daytime on dark nights and at night on brighter nights [12, 49]. Bobcats are described as having vision less suited for nighttime predation compared to other feline species [12, 28]. As such, they may take advantage of hunting during brighter periods but require greater vegetation cover for camouflage, while in contrast ocelots may rely on their superior vision to hunt effectively during darker nights. We observed no difference in step length of bobcats between lunar phase, contrasting prior studies [12, 49], however, we did observe differences in habitat selection of bobcats but not ocelots that may provide support for the visual acuity hypothesis.

Changing lunar conditions have the potential to influence the movement and habitat selection of nocturnal species [9, 16]. Past studies have examined activity levels of various carnivores in relation to moonlight, including ocelots and bobcats [17, 49, 53–55], as well as other mammals [8, 9, 11] using camera traps, however, our study provides the novel approach of also incorporating habitat selection in relation to lunar phase using high-frequency GPS data. In doing so, we provide evidence of differing responses to shifting lunar phase by two sympatric felids and show that ocelots alter step length but not habitat selection while bobcats altered habitat selection but not step length in response to shifting lunar phase. These differences between species may help to facilitate coexistence during shifting nocturnal environments (i.e., new moons versus full moons). Further, we provide some potential evidence for the visual acuity hypothesis, wherein ocelots, the more visually adept species increase movement during the darkest periods, potentially to mirror activity of prey. In contrast, bobcats, the more crepuscular species with less developed nocturnal vision [26, 28], showed greater selection for vegetation cover during brighter periods, potentially to facilitate hunting without detection by prey.

Identifying the abiotic factors that influence movement and habitat selection are essential to understanding the ecology of a species and can provide insight into interspecific coexistence, foraging ecology, and recovery efforts for endangered species such as the ocelot. Our results may also have applications for wildlife-road ecology studies, wherein lunar periods of higher movement may influence the timing and where species successfully cross roadways [33] or succumb to vehicle-collisions [32, 56–58]. In the case of ocelots, understanding the factors that influence movement can play a vital role in management by better identifying periods of higher movement which may influence ocelot-vehicle collisions, an important source of mortality for Texas ocelots [32]. As such, darker nights may be particularly high-risk for vehicle collisions due to decreased visibility and higher movement rates of ocelots.

## Acknowledgments

We would like to thank all of our technicians and collaborators on the project that made this study possible. This is manuscript number 098 of the East Foundation and manuscript number 23–105 of the Caesar Kleberg Wildlife Research Foundation.

## Author Contributions

**Conceptualization:** Maksim Sergeyev, Jason V. Lombardi, Michael E. Tewes.

**Data curation:** Maksim Sergeyev, Jason V. Lombardi.

**Formal analysis:** Maksim Sergeyev.

**Funding acquisition:** Michael E. Tewes, Tyler A. Campbell.

**Methodology:** Maksim Sergeyev, Jason V. Lombardi.

**Project administration:** Michael E. Tewes, Tyler A. Campbell.

Writing – original draft: Maksim Sergeyev.

Writing – review & editing: Maksim Sergeyev, Jason V. Lombardi, Michael E. Tewes, Tyler A. Campbell.

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
