## [Decision Letter · Decision Letter 0]

25 Apr 2023

PONE-D-23-04536Ocelots in the Moonlight: Influence of Lunar Phase on Habitat Selection and Movement of Two Sympatric FelidsPLOS ONE

Dear Dr. Sergeyev,

Thank you for submitting your manuscript to PLOS ONE. After careful consideration, we feel that it has merit but does not fully meet PLOS ONE’s publication criteria as it currently stands. Therefore, we invite you to submit a revised version of the manuscript that addresses the points raised during the review process.

Your manuscript has the potential to provide a useful contribution to the literature but requires revision. One of the primary publication criteria at PLOS is for the reader to be able to follow the authors through to their conclusions. As it reads now, there are some areas of the text that require clarity, others requiring additional details, and some where assumptions have been made that need to be explicit and the rationale provided. Two reviewers provide helpful comments below that will be useful in your revisions.

We look forward to receiving your revised manuscript.

Kind regards,

Stephanie S. Romanach, Ph.D.

Academic Editor

PLOS ONE

“Cooperative funding was provided by East Foundation, U.S. Fish and Wildlife Service, the Brown Foundation, the Tim and Karen Hixon Foundation and the Feline Research Program at the Caesar Kleberg Wildlife Research Institute.”

Reviewers' comments:

Reviewer's Responses to Questions

**Comments to the Author**

1. Is the manuscript technically sound, and do the data support the conclusions?

Reviewer #1: Yes

Reviewer #2: Yes

2. Has the statistical analysis been performed appropriately and rigorously? 

Reviewer #1: Yes

Reviewer #2: Yes

3. Have the authors made all data underlying the findings in their manuscript fully available?

Reviewer #1: No

Reviewer #2: No

4. Is the manuscript presented in an intelligible fashion and written in standard English?

Reviewer #1: Yes

Reviewer #2: Yes

5. Review Comments to the Author

Reviewer #1: Overall assessment:

The manuscript evaluates the effect of moon phase on habitat selection and movement of ocelots and bobcats in southern Texas, USA. Response of wildlife to moonlight is a growing area of study, and the effect of moonlight on wildlife behavior is still poorly understood for many species. This manuscript contributes to knowledge of carnivore response to moonlight. There are opportunities in the revision to clarify language and add additional citations. I have offered specific questions and comments below.

Specific comments:

Line 31: step selection function or SSF could also be useful keyword

Line 36: ‘across temporal scales’ or ‘lunar cycle’ to replace ‘diel cycle’ since focus of paper is on moon phases which cover multiple diel cycles, and movement is not assessed across diel cycle, only assessed during night

Line 45: Prugh and Golden 2014 (https://doi.org/10.1111/1365-2656.12148) would be a great meta-analysis to cite and discuss in this paragraph.

Line 81-82: Name the favored prey species

Line 101: There is an opportunity to clarify what is meant by the terms activity and movement. Specify the connections among step length, movement, and activity patterns more clearly and precisely throughout the manuscript. For example, Marneweck et al 2021 (https://doi.org/10.1002/ecy.3319) state: “movement rates represent step length (displacement) rather than overall activity (as measured in other studies), and heightened activity does not necessarily result in increased spatial displacement”.

Line 119: Easement land remains in private ownership and is a voluntary legal agreement that restricts habitat alteration/development by landowner. To say “owned by USFWS” is incorrect.

Line 129 - 130: Is there a seasonality to this rainfall?

Line 149: Would be helpful to break analysis into habitat selection and step length, to mirror how this is presented in results

Line 156: What distribution was specified?

Line 159: Classified into what? Can more detail be provided regarding this classification method? It would be helpful to include this habitat information (from lines 157 - 165) in a section prior to the step selection analysis

Line 161 - 163: None of these variables are highly correlated?

Line 167 - 168: Is it reasonable to assume no effect of season (ex. Temperature effect) and/or year?

Lines 170 - 178: Is it reasonable to assume that step length is not affected by habitat? Ex. as in Nisi et al. 2022 (https://doi.org/10.1111/1365-2656.13613).

Line 182: Can the authors provide more information about the collared individuals - sex, age, temporal coverage of collar period.

Line 189 & 200: What is α?

Line 205: What is the precision of GPS fix for collars used - is this difference between step length means greater than the expected GPS fix error?

Line 244 - 345: Here is an example where activity and movement are used synonymously, and the credibility and validity of this link should be supported and perhaps critiqued within the manuscript by the authors.

Tables 1 & 2: Does the standard error presented refer to the coef or exp(coef)?

Tables 1 & 2: Is there an intercept to report for these models? Also while not necessary the random effects could be reported.

Tables 1, 2, & 3: Present values with consistent decimals. Here 3 or 4 are used and in text 2 are used. Make consistent throughout manuscript

Reviewer #2: The authors present a wonderful case study showing that ocelots in southern Texas are more active under new moon conditions than full moon and that bobcat activity is not affected by lunar phase. Overall the study is very strong and bases its conclusions on the data they have collected. Although their approach is "simple", it is quite elegant and powerful. I found the manuscript well written and it is a much needed contribution to the field of understanding how lunar conditions affect/drive animal behavior and movement.

I really like this study and the manuscript. My only small suggestion is that the authors should mention what the light pollution is at their sites. they could use lightpollutioninfo.org and the new world atlas coupled with the preprint by Seymoure et al on SSRN on lunar equivalent light pollution to either say that these sites were not affected by light pollution or that some sites had light pollution which could have affected the cats' abilities to determine lunar illuminance.

Great study!

6. PLOS authors have the option to publish the peer review history of their article (what does this mean?). If published, this will include your full peer review and any attached files.

Reviewer #1: No

Reviewer #2: No

---

## [Author Response · Author response to Decision Letter 0]

10 May 2023

Thank you to the editors and reviewers for their comments on our manuscript. We have provided a detailed account of the changes made in our cover letter. We look forward to your reply!

---

## [Editor Report · Decision Letter 1]

15 May 2023

Ocelots in the Moonlight: Influence of Lunar Phase on Habitat Selection and Movement of Two Sympatric Felids

PONE-D-23-04536R1

Dear Dr. Sergeyev,

We’re pleased to inform you that your manuscript has been judged scientifically suitable for publication and will be formally accepted for publication once it meets all outstanding technical requirements.

Kind regards,

Stephanie S. Romanach, Ph.D.

Academic Editor

PLOS ONE

---

## [Editor Report · Acceptance letter]

22 Nov 2023

PONE-D-23-04536R1 

Ocelots in the Moonlight: Influence of Lunar Phase on Habitat Selection and Movement of Two Sympatric Felids 

Dear Dr. Sergeyev:

I'm pleased to inform you that your manuscript has been deemed suitable for publication in PLOS ONE. Congratulations! Your manuscript is now with our production department. 

Kind regards, 

on behalf of

Dr. Stephanie S. Romanach 

Academic Editor

PLOS ONE